# SEMANTIC ENERGY: DETECTING LLM HALLUCINATION BEYOND ENTROPY

## ABSTRACT

Large Language Models (LLMs) are being increasingly deployed in real-world applications, but they remain susceptible to hallucinations, which produce fluent yet incorrect responses and lead to erroneous decision-making. Uncertainty estimation is a feasible approach to detect such hallucinations. For example, semantic entropy estimates uncertainty by considering the semantic diversity across multiple sampled responses, thus identifying hallucinations. However, semantic entropy relies on post-softmax probabilities and fails to capture the model's inherent uncertainty, causing it to be ineffective in certain scenarios. To address this issue, we introduce Semantic Energy, a novel uncertainty estimation framework that leverages the inherent confidence of LLMs by operating directly on logits of penultimate layer. By combining semantic clustering with a Boltzmann-inspired energy distribution, our method better captures uncertainty in cases where semantic entropy fails. Experiments across multiple benchmarks show that Semantic Energy significantly improves hallucination detection and uncertainty estimation, offering more reliable signals for downstream applications such as hallucination detection. The code and intermediate data are available at https://anonymous.4open.science/submit4iclr.

## 1 INTRODUCTION

Large Language Models (LLMs) have been widely deployed in various aspects of production and daily life, demonstrating strong capabilities in different fields (Schlegel et al., 2025; Xiang et al., 2025). However, LLMs are still prone to being influenced by hallucinations and are prone to generate incorrect answers in situations where they lack knowledge, thus misleading users into making errors (Zhou et al., 2024; Farquhar et al., 2024). Recently, uncertainty estimation has been shown to be a reliable indicator for detecting hallucinations, reflecting the tendency of an LLM to generate hallucinations (Xiao & Wang, 2021; Huang et al., 2024). When the uncertainty of an LLM response is high, it often suggests a greater likelihood that the response is a hallucination, prompting further actions such as self-reflection (Renze & Guven, 2024; Kirchhof et al., 2025), regenerating of answers (Xu et al., 2025), or intervention by human experts (Liu et al., 2025; Hopkins et al., 2025).

Entropy is a commonly used metric for estimating uncertainty in LLM (Cheng et al., 2025; Duan et al., 2024). Similarly to traditional discriminative models, high entropy indicates high uncertainty because it means that the model cannot confidently select a particular outcome. However, due to the nature of natural language, the entropy of a single response cannot accurately reflect the reliability of LLMs. Specifically, even though LLMs may not confidently generate the next token, the semantic meaning of any generated token can still be the same. In such cases, we cannot identify an unreliable response simply attributing to its low probability of being generated. To accurately describe the uncertainty of responses composed of natural language, semantics must be considered.

Semantic entropy (Farquhar et al., 2024) is a typical method to characterize the semantic uncertainty of responses, effectively representing the probability that an LLM generates hallucinations. Given a question, semantic entropy involves sampling multiple responses, clustering them based on their semantic meaning, and then replacing individual responses with clusters to calculate entropy, thus achieving semantic-aware uncertainty characterization. Based on this method, a wide range of downstream applications have been developed, such as guiding Chain-of-Thought (CoT) reasoning (Ye et al., 2025) and parallel thinking (Xu et al., 2025). However, semantic entropy has

significant drawbacks stemming from entropy itself: it fails to capture the model's inherent uncertainty, leading to its ineffectiveness in some scenarios.

A representative case occurs when the model produces identical responses in multiple sampling instances for a given question, as illustrated in Fig. 1. According to semantic entropy, the resulting value is 0, which is considered a reliable response. However, even answering incorrectly, LLMs might also provide responses with the same semantics. Among samples with consistently semantic responses across multiple responses, the proportion of incorrect responses (like `Question3` in Fig. 1) approaches 50% in some datasets. In such cases, it is necessary to leverage the model's inherent uncertainty for differentiation: even if the LLM provides multiple responses with the same semantics for two different questions, their corresponding reliability still differs. In scenarios with a higher inherent uncertainty in the model, the likelihood of the LLM making mistakes is greater.

Several previous studies have shown that logits exhibit stronger inherent capabilities to characterize uncertainty compared to probabilities, and the magnitude of logits can indicate whether the model has undergone adequate training in a given scenario (Liu et al., 2020; Fu et al., 2025; Zhang et al., 2024). For example, in out-of-distribution (OOD) detection, studies have highlighted that the logit values for in-distribution (InD) samples are significantly higher than those for OOD samples (Liu et al., 2020). Recent work named LogToKU (Ma et al., 2025) points out that probabilities lose the intensity information of logits during normalization, thus limiting their ability to represent the inherent uncertainty of LLM. From this insight, we propose a new method to improve the failure cases of Semantic Entropy, termed *Semantic Energy*. Specifically, for a given prompt, we first perform multiple response samplings, followed by semantic sampling. When calculating the final uncertainty, rather than relying on probability as in Semantic Entropy, we estimate the response uncertainty based on logits, enabling the estimated uncertainty to reflect the model's inherent uncertainty. Our proposed metric significantly outperforms Semantic Entropy in evaluating the reliability of LLM responses, particularly in scenarios where Semantic Entropy fails. The main contributions are as follows:

- We expose the limitations of current uncertainty estimation methods based on probability and identify the failure cases in Semantic Entropy.
- We introduce **Semantic Energy**, a novel framework to evaluate the uncertainty of LLM responses, which indicates potential errors in the responses.
- We instantiate Semantic Energy using the Boltzmann formulation, and in the hallucination detection task, it achieves an average performance improvement of more than 13% compared in terms of AUROC to Semantic Entropy in cases where the latter is confident.

## 2 PRELIMINARIES

### 2.1 ESTIMATING LLM UNCERTAINTY WITH TOKEN-LEVEL ENTROPY

Let $q$ denote a natural language query provided as input to the LLM. Given the prompt $q$, a single response sequence is generated in an auto-regressive manner. This response can be represented as:

$$\boldsymbol{x} = [x_1, x_2, \ldots, x_T], \tag{1}$$

where $\boldsymbol{x}$ denotes a complete token sequence of variable length $T$. At each decoding step $t$, the model calculates a probability distribution across its entire vocabulary $\mathcal{V}$, assigning a conditional probability $p(x_t \mid x_{<t}, \boldsymbol{q})$ to each candidate token given the preceding context and the original query. To quantify the uncertainty in the LLM predictions at each generation step, the token-level entropy for position $t$ is formally defined as:

$$H_t = -\sum_{x \in \mathcal{V}} p(x \mid x_{<t}, \boldsymbol{q}) \log p(x \mid x_{<t}, \boldsymbol{q}), \tag{2}$$

where $\mathcal{V}$ is the model vocabulary. A higher value of $H_t$ indicates a more uniform probability distribution and thus greater uncertainty in the model choice for the $t$-th token.

To estimate the overall uncertainty associated with the entire generated response $\boldsymbol{x}$, a common and straightforward strategy is to aggregate these local token-level entropy values. The most common aggregation method is to compute the arithmetic mean across all tokens in the sequence:

$$H_{\text{avg}}(\boldsymbol{x}) = \frac{1}{T} \sum_{t=1}^{T} H_t, \tag{3}$$

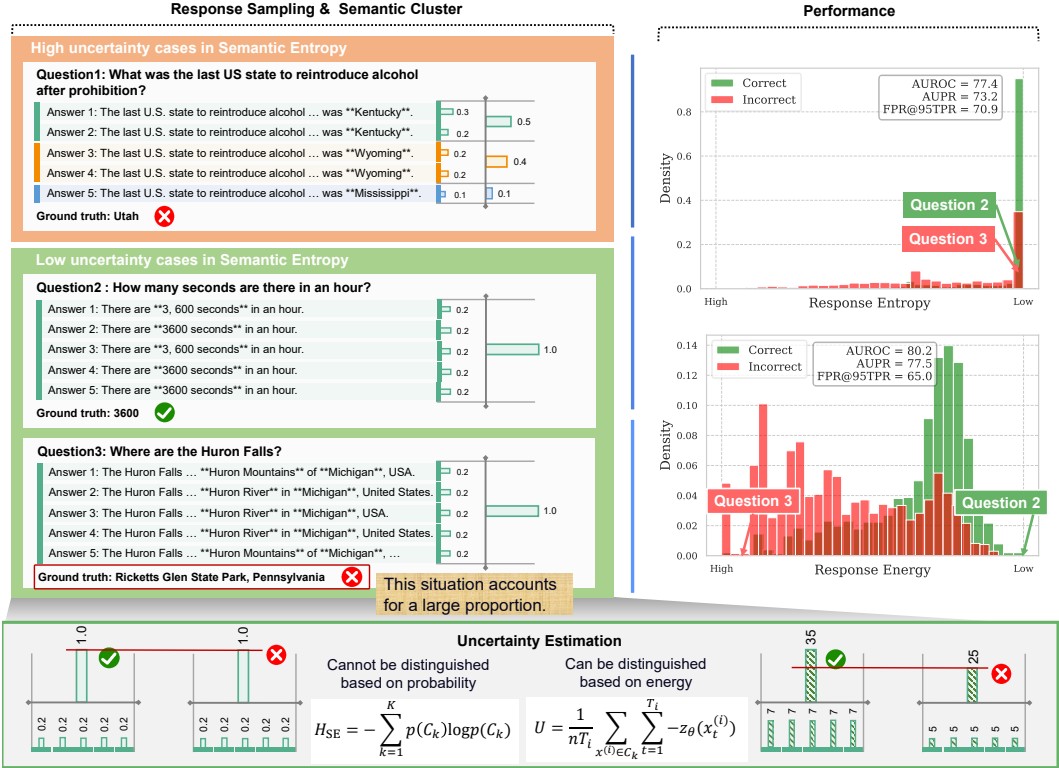

Figure 1: An intuitive comparison between Semantic Entropy and Semantic Energy in their ability to characterize uncertainty. Both approaches first sample and perform semantic clustering over distinct clusters of activity. The difference lies in the computation: Semantic Entropy is calculated based on normalized probabilities, while Semantic Energy is derived from logits. This enables Semantic Energy to distinguish cases when Semantic Entropy fails.

where $H_{\text{avg}}(\boldsymbol{x})$ serves as a proxy for the total uncertainty of the response, with a higher average entropy suggesting a more uncertain generation process. However, this approach implicitly assumes that each token contributes equally to the overall uncertainty, which may not hold in practice.

Recognizing that different tokens can carry varying levels of importance for the meaning and correctness of the final response, some recent studies (Duan et al., 2024) have proposed a refinement employing a weighted average. This method aims to amplify the contribution of critical or pivotal tokens (e.g., those conveying key facts or decisive information) to the final uncertainty score:

$$H_{\text{wavg}}(\boldsymbol{x}) = \sum_{t=1}^{T} w_t H_t, \quad \text{where} \sum_{t=1}^{T} w_t = 1, \quad (4)$$

where weights $w_t$ can be determined based on heuristic rules (such as focusing on entities) or learned mechanisms designed to identify semantically important tokens.

Although token-level entropy offers a fine-grained, local perspective on the uncertainty during the auto-regressive generation process, it possesses an inherent limitation: it operates purely on a syntactic or surface level. Quantifies the model's hesitation in choosing the next token, but does not necessarily reflect uncertainty over the underlying meaning or intent of the full response. This is *because vastly different token sequences can express the same semantic content, while highly similar token-level probability distributions might lead to responses with divergent meanings*. Consequently, token-level metrics may not fully capture the diversity in the semantic content of different possible responses. This critical shortcoming motivates the need for a more holistic, higher-level notion of uncertainty that operates on the distribution of semantically distinct outputs, leading to the concept of *semantic entropy* (Kuhn et al., 2024).

## 2.2 Semantic Entropy and Response Clustering

To capture semantic-level uncertainty, semantic entropy samples a set of $n$ candidate responses to the query $\boldsymbol{q}$ from LLM:

$$\mathbb{X} = \{\boldsymbol{x}^{(1)}, \boldsymbol{x}^{(2)}, \ldots, \boldsymbol{x}^{(n)}\}, \quad \boldsymbol{x}^{(i)} = [x_1^{(i)}, x_2^{(i)}, \ldots, x_{T_i}^{(i)}], \tag{5}$$

where $\boldsymbol{x}^{(i)}$ indicates the $i$-th sampled response and $T_i$ indicates the number of tokens in this response. Each response $\boldsymbol{x}^{(i)}$ has an associated likelihood in the model:

$$p(\boldsymbol{x}^{(i)} \mid \boldsymbol{q}) = \prod_{t=1}^{T_i} p(x_t^{(i)} \mid x_{<t}^{(i)}, \boldsymbol{q}), \quad \bar{p}(\boldsymbol{x}^{(i)}) = \frac{p(\boldsymbol{x}^{(i)} \mid \boldsymbol{q})}{\sum_{j=1}^{n} p(\boldsymbol{x}^{(j)} \mid \boldsymbol{q})}. \tag{6}$$

where $\bar{p}(\boldsymbol{x}^{(i)})$ indicates a normalized distribution over the sampled responses. Due to surface-level variability in language, semantically similar responses may have different forms. Therefore, semantic entropy clusters the responses into $K$ semantically coherent groups:

$$\mathbb{C} = \{\mathbb{C}_1, \mathbb{C}_2, \ldots, \mathbb{C}_K\}, \quad \mathbb{C}_k \subseteq \mathbb{X}, \tag{7}$$

where each cluster $\mathbb{C}_k$ contains responses that are semantically equivalent. The probability mass assigned to each cluster is defined as the sum over its members:

$$p(\mathbb{C}_k) = \sum_{\boldsymbol{x}^{(i)} \in \mathbb{C}_k} \bar{p}(\boldsymbol{x}^{(i)}). \tag{8}$$

Finally, *semantic entropy* ($H_{\text{SE}}$) is computed by applying the standard Shannon entropy formula to this distribution over semantic clusters:

$$H_{\text{SE}} = -\sum_{k=1}^{K} p(\mathbb{C}_k) \log p(\mathbb{C}_k), \tag{9}$$

which quantifies the model's uncertainty over distinct meanings conveyed by its responses. However, semantic entropy fails in some scenarios due to the limitations of entropy-based uncertainty estimation.

# 3 Modeling Uncertainty via Semantic Energy

## 3.1 Limitations of Entropy-Based Uncertainty Estimation

While $H_{\text{SE}}$ captures semantic variability, it only reflects *aleatoric uncertainty*—uncertainty arising from intrinsic randomness in the generation process. However, it fails to capture *epistemic uncertainty*—uncertainty stemming from the model's lack of knowledge. For example, as illustrated by the pair of instances in *Low uncertainty cases in Semantic Entropy* (see Fig. 1):

(1) Consider two queries $\boldsymbol{q}_2$ and $\boldsymbol{q}_3$, where the model has been extensively trained on data related to $\boldsymbol{q}_2$ (thus confident), but has limited exposure to $\boldsymbol{q}_3$ (thus uncertain).

(2) Assume that each query is sampled to obtain 5 responses, which were subsequently grouped into a certain cluster based on their semantic similarity ($K = 1$), respectively.

(3) In this case, $H_{\text{SE}} = 0$ for both, despite the LLM outputs 5 incorrect answers on $\boldsymbol{q}_3$ with the same semantic.

This means that if an LLM gives a wrong answer, and the semantics of multiple sampled results are all aligned with that wrong answer, then semantic entropy will mistakenly identify it as reliable. Unfortunately, LLMs are very good at "steadfastly" repeating the same wrong response. In some datasets, nearly half of the samples with repeated identical semantics are actually incorrect answers, making this limitation impossible to ignore.

The reason behind this is that the entropy calculated based on probabilities only reflects the relative likelihood of a particular LLM response compared to other possible responses generated by the

model, rather than the actual probability of that response to the question in the real world. However, when computing the probability of the next token, the model approximates the partition function as the sum of probabilities over its vocabulary. Therefore, for entropy to accurately represent uncertainty, two assumptions must hold: (1) the model has seen all possible responses (that is, the training distribution matches the real-world distribution perfectly), and (2) the model has fit the training distribution without bias (that is, the model output distribution matches the training distribution exactly). Clearly, neither of these assumptions holds. Worse still, the LLM response is based on the joint probability prediction of many tokens, rather than the single word classification of traditional discriminative models, which causes the error of the approximated partition function to accumulate to a degree that can no longer be ignored.

## 3.2 Energy-Based Confidence Estimation

To address this limitation, we introduce an energy-based formulation that complements semantic entropy and captures epistemic uncertainty. In thermodynamics of physics, lower energy corresponds to a more stable and less random state. Drawing on thermodynamic analogies, we treat lower-energy states as higher-confidence predictions, following the intuition that physical systems evolve toward minimal-energy configurations.

### 3.2.1 Boltzmann Distribution

The classical Boltzmann distribution defines the probability that a system occupying a state is capable of generating $x_t^{(i)}$ as:

$$p(x_t^{(i)}) = \frac{e^{-E_t^{(i)}/k\tau}}{Z_t}, \tag{10}$$

where $k$ is the Boltzmann constant, $\tau$ is the temperature, and $E_t^{(i)}$ is the token energy $x_t^{(i)}$, and $Z_t$ is the partition function. Specifically, for LLMs, $Z_t = \sum_{x \in \mathbb{V}} e^{-E_t(x)}$ is the normalization value across the entire vocabulary $\mathbb{V}$ (the difference between $\mathcal{V}$ and $\mathbb{V}$ is that $\mathcal{V}$ is the vocabulary in the predefined tokenizer of a specific LLM, while $\mathbb{V}$ is the space of possible next tokens in the real world, which is infinite and intractable). For simplicity, we assume that $Z_t$ is constant across $t$. The probability of a complete sequence and the average sequence-level energy can be represented as:

$$p(\boldsymbol{x}^{(i)}) = \prod_{t=1}^{T_i} p(x_t^{(i)}) = \frac{e^{-\sum_{t=1}^{T_i} E_t^{(i)}}}{\prod_{t=1}^{T_i} Z_t}, \quad E(\boldsymbol{x}^{(i)}) = \frac{1}{T_i} \sum_{t=1}^{T_i} E_t^{(i)}. \tag{11}$$

Suppose that we want to evaluate the total energy of a set $\mathbb{C}$. According to the Boltzmann equation, the total energy of $\mathbb{C}$ is the sum of its different states:

$$E_{\text{Bolt}}(\mathbb{C}) = \sum_{\boldsymbol{x}^{(i)} \in \mathbb{C}} E(\boldsymbol{x}^{(i)}). \tag{12}$$

Lower energy indicates that the cluster containing this response is more stable, i.e., it has lower uncertainty and thus higher reliability.

### 3.2.2 Specific Implementation in LLMs

For a LLM parameterized by $\boldsymbol{\theta}$ with a vocabulary $\mathcal{V}$, we can formulate the token-level energy distribution within the energy-based modeling framework. Specifically, each token $x_t^{(i)}$ is associated with an energy value $E(x_t^{(i)}, \boldsymbol{\theta})$, and the probability of generating this token is obtained via a Boltzmann distribution. The partition function $Z_{\boldsymbol{\theta}}$ serves as the normalization term, summing over all possible tokens in the vocabulary $\mathcal{V}$. This ensures that the resulting distribution is valid and comparable across tokens:

$$p(x_t^{(i)}, \boldsymbol{\theta}) = \frac{e^{-E(x_t^{(i)}, \boldsymbol{\theta})/k\tau}}{Z_{\boldsymbol{\theta}}}, \quad Z_{\boldsymbol{\theta}} = \sum_{x_t^{(i)} \in \mathcal{V}} e^{-E(x_t^{(i)}, \boldsymbol{\theta})/k\tau}, \tag{13}$$

where $k\tau$ corresponds to the temperature parameter that controls the sharpness of the distribution. The partition function $Z_{\boldsymbol{\theta}}$ reflects the dependence of the probability distribution on the model parameters $\boldsymbol{\theta}$.

From a Bayesian perspective, the total predictive uncertainty of the model should account for uncertainty in the parameters $\boldsymbol{\theta}$ themselves. This requires marginalizing over the posterior distribution of $\boldsymbol{\theta}$ given the training data $\mathcal{D}$. The marginal likelihood of generating a token $x_t^{(i)}$ under this treatment is expressed as:

$$p(x_t^{(i)}, \mathcal{D}) = \int_{\boldsymbol{\theta}} p(x_t^{(i)}, \boldsymbol{\theta}) \cdot \mathrm{p}(\boldsymbol{\theta} \mid \mathcal{D}) \, d\boldsymbol{\theta}. \tag{14}$$

However, we cannot obtain all possible $\boldsymbol{\theta}$, but we can only estimate uncertainty under specific models. For any given $\boldsymbol{\theta}$, the conditional probability of generating a token from a subset vocabulary $\mathcal{V} \subseteq \mathbb{V}$ is:

$$p(Z_{\boldsymbol{\theta},t} \mid Z_t) = \frac{\sum_{x_t^{(i)} \in \mathcal{V}} e^{-E(x_t^{(i)}, \boldsymbol{\theta})/k\tau}}{\sum_{x_t^{(i)} \in \mathbb{V}} e^{-E(x_t^{(i)}, \boldsymbol{\theta})/k\tau}}. \tag{15}$$

By combining this subset-based probability with Eq. 14, we can approximate the marginal distribution using the current model parameters. The resulting sampled approximation is given by:

$$\tilde{p}(x_t^{(i)}, \mathcal{D}) = p(x_t^{(i)}, \theta) \cdot p(Z_{\boldsymbol{\theta},t} \mid Z_t) = \frac{e^{-E(x_t^{(i)}, \boldsymbol{\theta})/k\tau}}{Z_t}, \tag{16}$$

where $\tilde{p}(x_t^{(i)}, \mathcal{D})$ indicates a sampled approximation of $p(x_t^{(i)}, \mathcal{D})$. Similarly to softmax-based probability modeling, the probability of a complete sequence is the joint probability of all tokens in the entire sequence:

$$\tilde{p}(\boldsymbol{x}^{(i)}) = \prod_{t=1}^{T_i} \tilde{p}(x_t^{(i)}) = \frac{e^{-\sum_{t=1}^{T_i} \tilde{E}_t^{(i)}/k\tau}}{\prod_{t=1}^{T_i} Z_t}, \quad \tilde{E}(\boldsymbol{x}^{(i)}) = \frac{1}{T_i} \sum_{t=1}^{T_i} \tilde{E}_t^{(i)}, \tag{17}$$

where $\tilde{E}(\boldsymbol{x}^{(i)})$ indicates the average sequence-level energy, which can also be interpreted as the energy per unit volume in thermodynamics, that is, the free energy density (Callen, 1993). To extend to semantic clusters, we treat each cluster $\mathbb{C}_k$ energy as a scaled joint energy:

$$\tilde{E}_{\mathrm{Bolt}}(\mathbb{C}_k) = \frac{1}{n} \sum_{\boldsymbol{x}^{(i)} \in \mathbb{C}_k} \tilde{E}(\boldsymbol{x}^{(i)}), \tag{18}$$

where $n$ are the sampling times for every question.

For an LLM trained with cross-entropy loss, we represent $E(x_t^{(i)}, \boldsymbol{\theta})$ as the negative value of the logit, that is, $E(x_t^{(i)}, \boldsymbol{\theta}) = -z_{\boldsymbol{\theta}}(x_t^{(i)})$, and $k\tau$ is by default the temperature used during LLM training, that is, $k\tau = 1$. The final uncertainty is defined as:

$$U(\boldsymbol{x}^{(i)}) = \frac{1}{n} \sum_{\boldsymbol{x}^{(j)} \in \mathbb{C}_k} \frac{1}{T_j} \sum_{t=1}^{T_j} -z_{\boldsymbol{\theta}}(x_t^{(j)}), \quad \forall \boldsymbol{x}^{(i)} \in \mathbb{C}_k, \tag{19}$$

which captures the average negative logit across tokens and samples. The lower energy corresponds to the lower uncertainty, thus establishing a direct connection between model confidence and energy-based representation. The uncertainty of a single reply is represented by the uncertainty of the semantic cluster to which it belongs. That is, for all replies that belong to the same semantic cluster, their uncertainty is identical.

## 4 EXPERIMENTS

### 4.1 SETUP

**Model & Baseline.** We conduct experiments using the Qwen3-8B model (Yang et al., 2025), and the ERNIE-21B-A3B model (MOE architecture) (Baidu, 2025). Our primary goal is to highlight the differences between probability-based methods and energy-based approaches. Therefore, we use the semantic entropy (Farquhar et al., 2024) as a baseline.

**Datasets & Metrics.** Experiments are performed on standard open-domain QA datasets in both Chinese and English: the Chinese dataset *CSQA* (He et al., 2024) and the English dataset *TriviaQA* (Joshi et al., 2017). To assess whether the estimated uncertainty can capture the risk that the model makes errors, we estimate the AUROC between uncertainty scores and correctness (i.e., whether the answer is correct).

## 4.2 MAIN RESULTS

Table 1: Uncertainty estimation performance on OpenQA Datasets.

| Model | Dataset | Semantic Entropy | | | Semantic Energy | | |
|---|---|---|---|---|---|---|---|
| | | AUROC | AUPR | FPR95 | AUROC$_{(\uparrow)}$ | AUPR$_{(\uparrow)}$ | FPR95$_{(\downarrow)}$ |
| **Qwen3-8B** | CSQA | 71.6% | 53.6% | 77.0% | 76.1% ⇑$_{4.5\%}$ | 61.4% ⇑$_{7.8\%}$ | 74.6% ⇑$_{2.4\%}$ |
| | TriviaQA | 69.6% | 73.5% | 79.1% | 74.8% ⇑$_{5.2\%}$ | 79.2% ⇑$_{5.7\%}$ | 74.7% ⇑$_{4.4\%}$ |
| **ERNIE-21B-A3B** | CSQA | 77.4% | 73.2% | 70.9% | 80.2% ⇑$_{2.8\%}$ | 77.5% ⇑$_{4.3\%}$ | 65.0% ⇑$_{5.9\%}$ |
| | TriviaQA | 75.1% | 85.0% | 69.9% | 81.0% ⇑$_{5.9\%}$ | 89.9% ⇑$_{4.9\%}$ | 63.7% ⇑$_{6.2\%}$ |

Table 1 summarizes the performance of uncertainty estimation methods on the CSQA and TriviaQA datasets. We evaluate models using standard metrics: AUROC, AUPR, and FPR@95, based on whether the uncertainty score can discriminate correct from incorrect responses.

In both models and datasets, *semantic energy* consistently outperforms *semantic entropy*. On CSQA, the Boltzmann energy improves AUROC from 71.6% to 76.1% on Qwen3-8B and from 77.4% to 80.2% on ERNIE-21B-A3B. Similar trends are observed on TriviaQA, where Boltzmann energy yields AUROC gains of more than 5% compared to the semantic entropy. Improvements are also reflected in AUPR and FPR@95, indicating better calibration and reduced false positive rates.

These results highlight the robustness of energy-based uncertainty estimation, particularly in low-diversity scenarios where entropy becomes degenerate (details in Table 2). By incorporating internal model states via logits, semantic energy captures a richer signal for uncertainty estimation beyond probability-based entropy.

## 4.3 ABLATION STUDIES

### 4.3.1 RESULTS ON QUESTIONS WITH SINGLE CLUSTER

Table 2: Uncertainty estimation performance on questions with **single** cluster.

| Model | Dataset | Semantic Entropy | | | Semantic Energy | | |
|---|---|---|---|---|---|---|---|
| | | AUROC | AUPR | FPR95 | AUROC$_{(\uparrow)}$ | AUPR$_{(\uparrow)}$ | FPR95$_{(\downarrow)}$ |
| **Qwen3-8B** | CSQA | 50.0% | 55.8% | 95.0% | 66.7% ⇑$_{16.7\%}$ | 67.6% ⇑$_{11.8\%}$ | 80.3% ⇑$_{14.7\%}$ |
| | TriviaQA | 50.0% | 75.1% | 95.0% | 62.1% ⇑$_{12.1\%}$ | 81.6% ⇑$_{6.5\%}$ | 86.9% ⇑$_{8.1\%}$ |
| **ERNIE-21B-A3B** | CSQA | 50.0% | 77.0% | 95.0% | 58.9% ⇑$_{8.9\%}$ | 81.9% ⇑$_{4.9\%}$ | 88.4% ⇑$_{6.6\%}$ |
| | TriviaQA | 50.0% | 85.9% | 95.0% | 65.8% ⇑$_{15.8\%}$ | 91.4% ⇑$_{5.5\%}$ | 83.4% ⇑$_{11.6\%}$ |

In Table 2, we present the case where all responses share the same semantics, that is, all responses are clustered into a single group as described in Sec. 3.1. In this scenario, semantic entropy completely fails, whereas semantic energy is still able to provide a certain level of distinction, resulting in semantic energy achieving an average performance improvement of more than 13% compared in terms of AUROC to semantic entropy in cases where the latter is confident. It is important to note that the value of semantic entropy in such cases is always zero, meaning that its performance reflects the expected performance when the uncertainty indicator is meaningless, for example, the AUPR corresponds to the number of positive samples (i.e. correct responses).

### 4.3.2 ADVANTAGES OF SEMANTIC CLUSTER

Inspired by semantic entropy, we incorporate semantic influence when computing energy. If semantics are not considered and the energy of a single response is directly used to characterize the

reliability of an LLM's reply, such as in LogTokU (Ma et al., 2025), a clear problem arises: a single response having high energy does not necessarily mean that the entire cluster of responses sharing the same semantics also has high energy. This is because different responses belonging to the same semantic cluster can still have varying energy values. Therefore, we represent the energy of a response by the energy of the cluster to which it belongs. As shown in Fig. 2, we conduct an ablation study on whether to include semantics. The experimental results demonstrate that incorporating semantics significantly improves the accuracy of uncertainty estimation.

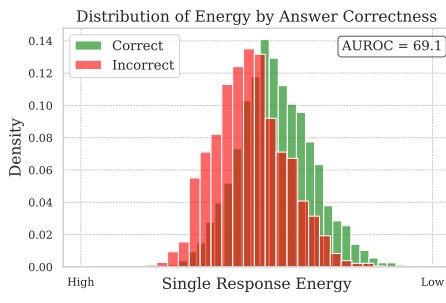

(a) TriviaQA (w/o semantic)

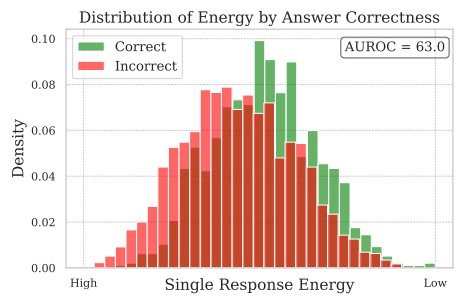

(b) CSimpleQA (w/o semantic)

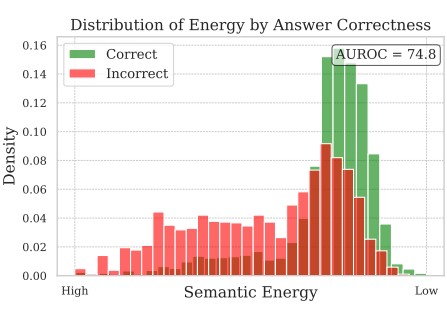

(c) TriviaQA (with semantic)

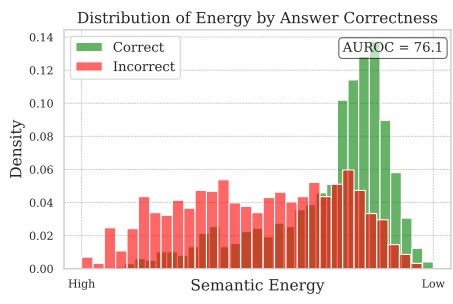

(d) CSimpleQA (with semantic)

Figure 2: Comparison of semantic vs. non-semantic uncertainty modeling on TriviaQA and CSimpleQA datasets.

### 4.3.3 RESULTS ON THINK MODE

To validate the performance of the proposed method on the thinking model, we conduct experiments on Qwen-8B using the CSQA dataset and explore the case where the think mode is enabled. Specifically, we activate the think mode but discard the content within `<think>...<think>` during evaluation, considering only the response portion. The final results, shown in Fig. 3, are consistent with the observations in Table 1. This indicates that even when the LLM undergoes a lengthy thinking process during output generation, its final results can still accurately capture the uncertainty of the model's responses through logits, thereby reflecting the reliability of the answers. Additionally, we observe that both semantic entropy and semantic energy demonstrate significantly improved uncertainty characterization capabilities in the `think` mode compared to the performance reported in Table 1. This suggests that the context during the deep thinking process may positively contribute to characterizing the distributional uncertainty of the final responses.

## 5 RELATED WORK

**Uncertainty estimation methods.** Recently, numerous uncertainty estimation methods for LLMs have been proposed. These include methods that utilize natural language for uncertainty feedback, including heuristically designed and trained approaches (Tao et al., 2025; Xiong et al., 2023; Lin et al., 2023); methods that estimate uncertainty based on model states, including those leveraging

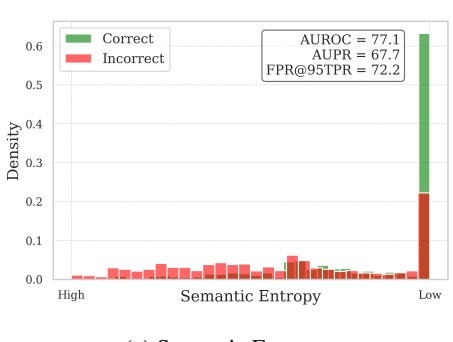 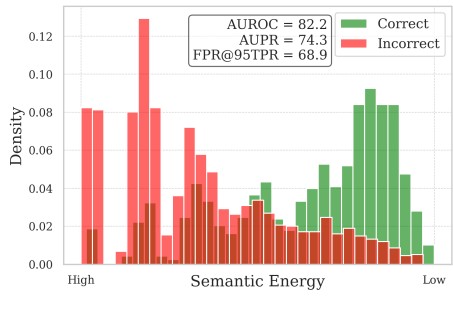

(a) Semantic Entropy            (b) Semantic Energy

Figure 3: Comparison of semantic entropy vs. semantic energy on CSQA datasets with think mode on.

prior knowledge or statistical observations of model states (Kostenok et al., 2023; Li et al., 2025; Liu et al., 2024), or observing changes under perturbations (Zhang et al., 2025b; Gao et al., 2024); and methods that take into account the semantics of the response, including consistency-based uncertainty characterizations (Lyu et al., 2025; Bartsch et al., 2023; Xiao et al., 2025) and approaches that integrate semantics with model states (Kuhn et al., 2024; Grewal et al., 2024).

**Uncertainty-guided applications.** The utilization of uncertainty estimation is widely applied in both the post-train and inference phases of LLMs. For example, minimizing entropy during the reinforcement learning process helps reduce uncertainty (Zhang et al., 2025a; Agarwal et al., 2025) and encourages exploration of critical positions with higher uncertainty (Zheng et al., 2025; Cheng et al., 2025). In the inference phase, uncertainty has emerged as a powerful signal for guiding LLMs and related systems. For example, Ye et al. (2025) introduce CoT Entropy to quantify the uncertainty of a PRM in stepwise verification, while Wang et al. (2022) demonstrate that monitoring uncertainty across multiple reasoning paths helps select more reliable outputs. In retrieval-augmented generation (RAG). Guo et al. (2025) propose empowering retrieval decisions with model confidence, and Chen & Varoquaux (2025) provide *internal confidence* to improve RAG on factual QA and mathematical reasoning tasks. In multi-agent and collaborative systems, Dey et al. (2025) propose *uncertainty-aware fusion* to reduces hallucinations by strategically combining multiple LLM based on their accuracy and self-assessment abilities, Kruse et al. (2025) propose *multi-LLM uncertainty via subset ensembles* that uses Jensen-Shannon Divergence to identify and aggregate well-calibrated subsets of LLMs. Uncertainty can also be used to determine when to stop or skip reasoning. Xu et al. (2025) design adaptive stopping criteria where the model halts reasoning once confidence exceeds a threshold, reducing unnecessary computation, and Zhu et al. (2025) propose *UnCert-CoT* that uncertainty-aware skipping prevents overthinking by allowing the model to bypass low-value reasoning steps.

## 6 DISCUSSION AND CONCLUSION

In this paper, we introduce the concept of semantic energy as an enhancement of semantic entropy, an uncertainty modeling method that substitutes entropy with energy (derived from logits). Semantic energy can effectively compensate for the shortcomings of semantic entropy and better capture the inherent uncertainty within models. We also clarify the limitations imposed by probability normalization and demonstrate the potential to overcome these constraints. Although logits are not strictly equivalent to energy, they exhibit energy-like characteristics solely due to the implicit constraints arising from network initialization and regularization during training. This property of logits has been leveraged in many previous studies, such as in the field of OOD detection. Furthermore, future LLM development should consider the limitations associated with probability-based training to mitigate performance degradation caused by factors such as training data distribution.

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

## A    PROMPTS FOR DIFFERENT EXPERIMENTS

---
**Prompt for Response Sampling**

**According to official recommendations, we adopted the following prompts for Qwen3 and ERNIE respectively.**

**Qwen-3 Prompt:**
```
<|im_start|>user\n {question} Give a short answer:
<|im_end|>\n <|im_start|>assistant\n
```

**ERNIE-4.5 Prompt:**
```
<|begin_of_sentence|>User: {question} Give a short answer:\n
Assistant:
```
---

---
**Prompt for Semantic Cluster**

**We utilize the `TIGER-Lab/general-verifier` model for semantic clustering, which analyzes whether different answers convey the same meaning for a given question.**

```
<|im_start|>system Please reason step by step, and put your
final answer within \\boxed.<|im_end|>\n <|im_start|>user\n
{question} \n\n{answer_a} \n\n{answer_b} \n\nFor the above
question, please verify if the student's answer is equivalent
to the ground truth answer.\nDo not solve the question by
yourself; just check if the student's answer is equivalent to
the ground truth answer.\nIf the student's answer is correct,
output "Final Decision:  Yes".  If the student's answer
is incorrect, output "Final Decision:  No".  Assistant:
<|im_end|>\n <|im_start|>assistant
```
---

---
**Prompt for Judgment**

**During the judgement process, `TIGER-Lab/general-verifier` is employed to determine whether the model's response aligns with the ground truth answer.**

```
<|im_start|>system Please reason step by step, and put your
final answer within \\boxed.<|im_end|>\n <|im_start|>user\n
{question}\n\n{llm_response}\n\n{ground_truth}\n\nFor the above
question, please verify if the student's answer is equivalent
to the ground truth answer.\nDo not solve the question by
yourself; just check if the student's answer is equivalent to
the ground truth answer.\nIf the student's answer is correct,
output "Final Decision:  Yes".  If the student's answer
is incorrect, output "Final Decision:  No".  Assistant:
<|im_end|>\n <|im_start|>assistant
```
---

## B    DETAILS FOR COMPARISON METHODS

In this paper, we primarily compare our method with the well-known approach, semantic entropy. Since both methods require response sampling and semantic clustering, we use identical data for these parts. That is, the only difference lies in the final uncertainty calculation process, in which the responses and clusters are generated from the same set of data. For the TriviaQA dataset, due to its large number of entries, we only estimate results for the first 5,000 samples. Note that this is a commonly adopted practice. Additionally, the sampling temperature is set to 0.6, as recommended

in the official documentation, and the random seed is set to values from 1 to 10 to sample ten distinct responses.

It is important to note that the value of semantic entropy in table 2 is always zero, meaning that its performance reflects the **expected** result when the uncertainty indicator is meaningless; for example, the AUPR corresponds to the number of positive samples (i.e. correct responses).

## C  PSEUDO CODE

---

**Algorithm 1:** *Semantic Energy*–based Uncertainty Estimation

---

**Input:** Natural language query $\boldsymbol{q}$, LLM with parameters $\boldsymbol{\theta}$, sampling times $n$
**Output:** Semantic clusters $\mathbb{C} = \{\mathbb{C}_1, \ldots, \mathbb{C}_K\}$ and their uncertainties $U(\mathbb{C}_k)$

1 **Step 1: Response Sampling**
2 **for** $i = 1$ **to** $n$ **do**
3     Sample response $\boldsymbol{x}^{(i)} = [x_1^{(i)}, \ldots, x_{T_i}^{(i)}]$ from LLM;
4     **for** $t = 1$ **to** $T_i$ **do**
5        Record logit $z_{\boldsymbol{\theta}}(x_t^{(i)})$;
6        Record probability $p(x_t^{(i)} \mid x_{<t}^{(i)}, \boldsymbol{q})$;
7     **end**
8 **end**
9 **Step 2: Semantic Clustering**
10 Initialize $\mathbb{C} \leftarrow \emptyset$;
11 **foreach** *pair* $(\boldsymbol{x}^{(i)}, \boldsymbol{x}^{(j)})$ **do**
12     Use semantic verifier to test equivalence;
13     **if** *equivalent* **then**
14        Assign $\boldsymbol{x}^{(i)}, \boldsymbol{x}^{(j)}$ to the same cluster;
15     **end**
16 **end**
17 Obtain $\mathbb{C} = \{\mathbb{C}_1, \ldots, \mathbb{C}_K\}$;
18 **Step 3: Energy-Based Reliability**
19 **foreach** *response* $\boldsymbol{x}^{(i)}$ **do**
20     Compute average energy:
$$\tilde{E}(\boldsymbol{x}^{(i)}) = \frac{1}{T_i} \sum_{t=1}^{T_i} -z_{\boldsymbol{\theta}}(x_t^{(i)})$$
21 **end**
22 **foreach** *cluster* $\mathbb{C}_k$ **do**
23     Compute cluster uncertainty:
$$U(\mathbb{C}_k) = \frac{1}{|\mathbb{C}_k|} \sum_{\boldsymbol{x}^{(i)} \in \mathbb{C}_k} \tilde{E}(\boldsymbol{x}^{(i)})$$
24 **end**
25 **foreach** *response* $\boldsymbol{x}^{(i)} \in \mathbb{C}_k$ **do**
26     Assign uncertainty $U(\boldsymbol{x}^{(i)}) = U(\mathbb{C}_k)$;
27 **end**

---

## D  THE USE OF LLMS

After completing the paper, we used an LLM to check for grammatical errors in the text, thereby ensuring that the paper was free of writing issues. At the same time, we continuously submitted updated versions to different LLMs, such as ChatGPT, for simulated review, which helped supple-

ment additional experiments and related work analyses. It is worth noting that although our final version satisfied the LLMs, our goal was not to make the LLMs happy with the revisions. Instead, we engaged in additional discussions and made modifications based on the potential shortcomings pointed out by the LLMs, continually refining the paper into its final form. While ensuring it meets the approval of peers, we also made sure the LLM considered this paper an outstanding piece of work.

# E    ATTEMPTS ON THE FERMI-DIRAC DISTRIBUTION

## E.1    FERMI-DIRAC DISTRIBUTION

To account for potential dependencies among samples, we generalize to the expected energy:

$$E(\mathbb{C}_k) = \sum_{\boldsymbol{x}^{(i)} \in \mathbb{C}_k} p(\boldsymbol{x}^{(i)}) E(\boldsymbol{x}^{(i)}). \tag{20}$$

Since exact $\boldsymbol{Z}$ is intractable in Boltzmann distribution, we also consider the **Fermi–Dirac distribution**:

$$p(\boldsymbol{x}^{(i)}) = \frac{1}{e^{(E(\boldsymbol{x}^{(i)})-\mu)/kT} + 1}, \tag{21}$$

where $\mu$ (chemical potential) is approximated as the mean of all logits across tokens and samples. This form reflects saturation effects and confidence plateauing.

The corresponding energy is defined as:

$$E_{\text{Fermi}}(\mathbb{C}_k) = \sum_{\boldsymbol{x}^{(i)} \in \mathbb{C}_k} \frac{E(\boldsymbol{x}^{(i)})}{e^{(E(\boldsymbol{x}^{(i)})-\mu)/kT} + 1}. \tag{22}$$

## E.2    HYPER-PARAMETERS IN FERMI-DIRAC DISTRIBUTION

In the Fermi–Dirac-based uncertainty formulation, the chemical potential $\mu$ plays a critical role in shaping the distribution and consequently in estimating model confidence. To investigate its effect, we empirically examined how the quality of uncertainty estimation varies with different choices of $\mu$. When $\mu$ is initially set to match the Boltzmann distribution regime (that is, very negative values), the Fermi–Dirac model behaves similarly to the exponential Boltzmann case. As $\mu$ increases, the performance first improves and reaches a peak, after which it drops rapidly, indicating a sharp sensitivity to this parameter.

From a physical perspective, the chemical potential $\mu$ in the Fermi–Dirac distribution determines the energy level at which the probability of occupation is $1/2$. In our setup, we interpret $\mu$ as a learned threshold separating high-confidence and low-confidence generations. To avoid manual tuning and leverage the thermodynamic grounding of the method, we solve for the optimal $\mu$ analytically by enforcing the self-consistency condition.

$$\frac{1}{n} \sum_{i=1}^{n} \frac{1}{e^{(E(\boldsymbol{x}^{(i)})-\mu)/kT} + 1} = \mu, \tag{23}$$

which is derived from the condition that the mean value of the Fermi-Dirac occupation function is equal to $\mu$ itself. This fixed-point equation,

$$\mathbb{E}_{\boldsymbol{x} \sim \mathbb{X}} \left[ \frac{1}{e^{(E(\boldsymbol{x})-\mu)/kT} + 1} \right] = \mu,$$

yields the *system-consistent* value of $\mu$, ensuring that the modeled uncertainty distribution reflects a stable equilibrium in the energy landscape. Numerically, we solve Eq. 23 using root-finding methods (e.g., bisection or Newton-Raphson), producing an interpretable and data-adaptive setting of $\mu$ without requiring heuristic tuning. However, these observations are not consistently manifested in all models and require further exploration.

