# OpenReview forum: "Semantic Energy: Detecting LLM Hallucination Beyond Entropy"
_ICLR.cc/2026/Conference — Submitted to ICLR 2026_

### Official Review · Reviewer_DB7j · 2025-10-21

**Soundness:** 2
**Presentation:** 2
**Contribution:** 2
**Rating:** 2
**Confidence:** 4

**Summary:**

This paper proposes semantic energy, an alternative to semantic entropy for uncertainty quantification in open-response QA.

I think the idea shows promise, but the paper needs more work, both in experimentation and exposition. For exposition, it is not clear exactly how their estimator is calculated. For experimentation, the results are only on 2 datasets with 2 models.

**Strengths:**

* UQ for LLMs is an area of growing interest
* The authors show improved performance on two datasets, using two models

**Weaknesses:**

* Lack of experiments: The paper only evaluates on two models, using two LLMs, and only compares against (one variant of) semantic entropy. I understand that not all researchers have the same access to resources... but this isn’t nearly enough to evaluate whether semantic energy outperforms semantic entropy. Further, I disagree that it is “sufficient” to compare just with semantic entropy, particularly with so few experiments — it is hard to know whether these are cases where semantic entropy does particularly poorly vs logit-based approaches such as mean token entropy, sequence probability, or perplexity. See under “questions” for more.
* Lack of experimental details:
    * There are a number of implementations of semantic entropy, some of which use logits, others use sample counts. Which is being used?
    * How many samples are used for clustering? What method is used for clustering (deberta, LLM-as-a-judge, something else?)
    * The authors mention that in some datasets, half of questions with zero estimated semantic entropy are wrong. I expected to see some numbers for the datasets in question.
    * In the single-cluster results, what is the accuracy?
    * For the single-cluster results, how are you calculating FPR@TPR=0.95? I assume you are either including all observations, no observations, or randomly sampling 95% of the data to get a TPR of 95%. But in either of these scenarios, I am surprised to see a FPR@TPR=0.95 of exactly 95%.


* Clarity: The section on the estimator is not very clear:
    * I wasn’t sure what the part about the uncertainty in theta refers to. We are working with a non-Bayesian model, so there is no uncertainty about the model parameters.
    * $Z_{\theta, t}$ is not defined, and it is not clear what the subset $\mathcal{V}$ is (or, how it relates to uncertainty about the model parameters)
    * Eq 19 doesn’t make sense — it only considers the kth cluster, and doesn’t indicate how we combine multiple clusters. As a result, I am unsure as to what the authors’ estimator actually is, and how it combines the energies/log likelihoods for the individual clusters.

**Questions:**

* I would like to see the authors better describe their estimator -- this would allow me to better evaluate the paper, and I could revisit my review with greater understanding of the methodology
* In terms of experiments, it would be nice to see a more thorough consideration of when semantic energy works, and why semantic energy works. For the former, the authors really need more datasets and models, plus more comparison methods. Some quesitons I would like to see answered are:
    * does semantic energy work well in cases where other logit-based methods like perplexity or mean token entropy work well?
    * as we increase the number of samples, do semantic entropy and semantic energy converge to the same value? (I think we would expect this).
    * Related to the above: The authors argue that the reason for energy working better is due to logits carrying more information about uncertainty than the probabilities. It is not clear that to me that this is the limiting factor here.

---

### Official Review · Reviewer_Podw · 2025-10-29

**Soundness:** 2
**Presentation:** 3
**Contribution:** 3
**Rating:** 6
**Confidence:** 4

**Summary:**

The paper introduces Semantic Energy, a new uncertainty estimation method for Large Language Models (LLMs). The method aims to overcome the limitations of Semantic Entropy (Farquhar et al., 2024), which relies on post-softmax probabilities and therefore cannot capture the model’s inherent uncertainty. Instead, the authors propose operating directly on the logits of the penultimate layer and defining an energy-based uncertainty score inspired by the Boltzmann distribution. By integrating semantic clustering with this logit-based energy formulation, the approach intends to capture both semantic and epistemic uncertainty. Experiments on Qwen3-8B and ERNIE-21B-A3B models, using TriviaQA and CSQA, show consistent improvements in AUROC and AUPR for hallucination detection. The paper also presents ablations (e.g., single-cluster cases, semantic vs. non-semantic variants) and extends the idea to a Fermi-Dirac formulation.

**Strengths:**

1. The motivation is clear and well connected to a real shortcoming of Semantic Entropy.
2. The derivation of the energy-based formulation is conceptually elegant and mathematically consistent.
3. The paper is very well written and easy to follow.
4. Experiments show consistent improvements over a strong baseline (Semantic Entropy), across two models and two datasets.
5. The ablations are thorough and empirically convincing.
6. The Fermi-Dirac extension, though exploratory, demonstrates the authors’ awareness of alternative formulations of the method.

**Weaknesses:**

1. The empirical scope is relatively narrow: only two models and two QA datasets are tested. While both are multilingual, the generalization to other domains (e.g., reasoning, dialogue, factuality) remains unclear.
2. Comparisons are limited to Semantic Entropy; other strong baselines such as Logit-based OOD detectors, Semantic Uncertainty (Kuhn et al., 2024), Sample Consistency (Lyu et al., 2025), IDK-token (Cohen et al. 2024), or Self-Reflective Uncertainties (Kirchhof et al., 2025) are missing.
3. Computational cost is not discussed — there's no significant computational gap comparing to semantic entropy of course, however, the proposed approach still requires multiple response samplings and semantic clustering, which can be expensive for large-scale applications.
4. The “think mode” experiment is interesting but underexplored — it would be valuable to clarify how the model’s internal reasoning sequence affects the uncertainty estimation beyond empirical observation.

**Questions:**

1. How sensitive is the method to the sampling temperature and the number of samples? Did you try higher temperatures or fewer samples to see how stable Semantic Energy remains?
2. Did you compare your approach with LogToKU (Ma et al., 2025) directly on the same datasets? The methods seem conceptually related.
3. Can the Fermi-Dirac variant be quantitatively compared to the Boltzmann version (e.g., on AUROC/AUPR) to justify its inclusion beyond theoretical curiosity?
4. Did you consider evaluating on datasets that measure long-form generation (e.g., HaluBench or FEVER) to assess scalability and context length sensitivity?

---

### Official Review · Reviewer_FSzi · 2025-11-06

**Soundness:** 1
**Presentation:** 1
**Contribution:** 2
**Rating:** 2
**Confidence:** 4

**Summary:**

Authors propose combining the core idea of "semantic entropy" (computing entropy after clustering different generations by their semantic meaning) with the idea of using "energy" (i.e. using logits before they're normalized into logprobs with a softmax) as a measure of uncertainty in natural-language generation QA tasks.
Authors claim empirical improvements in UQ performance of Semantic Energy over Semantic Entropy.

**Strengths:**

S1. I find the core idea (as described in the summary) interesting (but incremental) and I think it's worthwhile effort to evaluate it.

**Weaknesses:**

W1. Incremental contribution.
The core idea is incremental, given existence of Semantic Entropy (SE) (Kuhn et al.), and LogTokU (Ma et al., 2025), which uses logits as UQ-score in NLG settings (albeit the evaluation in that paper is also rather lacking).

W2. Lacking appropriate empirical comparison to prior work.

W2a. More methods.
The key results in Tables 1 and 2 should incorporate at the very least LogTokU, this key ablation cannot be conducted just in the form of Figure 2.
I'd particularly like to see a comparison to KLE (Nikitin et al., 2024) which based on other papers in my batch seems to be one of the SOTA extensions of SE.

W2b. More models.

W2c. More datasets.

W2d. Confidence intervals.

W2e. There are codebases available which would make the process rather straightforward, given the method proposed is rather simple to implement once the Semantic-Clustering is computed: https://github.com/AlexanderVNikitin/kernel-language-entropy


W3. Vague (and at times incorrect) discussion/description disguised in technical terminology.

W3a. e.g. "fails to capture the model's inherent uncertainty", "leverages the inherent confidence".

W3b. L198-204: The discussion of aleatoric and epistemic uncertainty and assumed causation in (1) "limited exposure" is not supported by either citations or empirical evidence. One could argue that SE captures a combination of both aleatoric and epistemic uncertainty - how would authors respond to this?

W3c. L224: "the error of the approximated partition function" - is there a "true value of the partition function"? How is it defined?

W3d. Section 3.2.

W3da. I understand that $\mathcal{V}$ is the token dictionary. I don't understand what is $\mathbb{V}$ - could the authors explain? It seems to me that contemporary tokenizers can tokenize any sequence of strings we would like to model.

W3db. eq 13 - authors surely don't mean a joint probability distribution over the token x_t^i and the parameters of the LLM.

W3dc. To my best judgement, the mathematics of eqs 14-16 is simply incorrect. I don't see any way how it could be derived. Some of it can be fixed, but I think the entire discussion of Sec 3.2.2 is superfluous and trying to add mathematical rigour to motivation that is driven by analogy that is not strictly speaking riguorous. Authors can find similar exposition of the motivation (but in my opinion correct) e.g. in the by now "reference citation" of Liu et al., 2020, which authors cite. Ultimately, eq 19 doesn't really depend on anything that came before and that's the UQ-score/estimator proposed and used in Section 4.

**Questions:**

See Weaknesses.

Typos:
- L074 "semantic sampling" -> "semantic clustering"

---

> ### Author Response · Authors · 2025-11-13
> **Can you provide a professional explanation to support your point?**
>
> You claim that Eqs. 14–16 are incorrect, but your reasoning is based on "to my best judgment" and "simply incorrect". Such descriptions appear highly unprofessional in an academic discussion. If you believe there is an error in the mathematical derivation, you should specify what the correct derivation should be, or at least provide a disproof.
>
> We look forward to your reply to clarify our doubts. Thank you.

---

> ### Comment · Reviewer_FSzi · 2025-11-13
>
> I used the statement "to my best judgment" because it leaves the doors open for the authors to try to convince me otherwise if they feel I'm wrong.
>
> However, generally, I agree with you, I could've spelled out the mistakes straight away, apologies.
>
> W3dca. L267: "marginal likelihood": $p(x^{(i)}_t, D)$ is not marginal likelihood, it's a joint probability distribution.
>
> W3dcb. eq (14): $\int_{\boldsymbol{\theta}} p(x_t^{(i)}, \boldsymbol{\theta}) p(\boldsymbol{\theta} \mid \mathcal{D})  d\boldsymbol{\theta}$ does not form a straightforward joint distribution and so cannot be easily marginalized.
>
> I'm guessing the authors meant (fixing W3dca and W3dcb):
> Assuming $X_t^{(i)}$ is independent of $D$ when conditioned on $\boldsymbol{\theta}$, $p(x_t^{(i)} | \boldsymbol{\theta}) = p(x_t^{(i)} | \boldsymbol{\theta}, D)$, which is a standard assumption.
> $\int_{\boldsymbol{\theta}} p(x_t^{(i)} | \boldsymbol{\theta}) p(\boldsymbol{\theta} \mid \mathcal{D})  d\boldsymbol{\theta} = \int_{\boldsymbol{\theta}} p(x_t^{(i)} | \boldsymbol{\theta}, D) p(\boldsymbol{\theta} \mid \mathcal{D})  d\boldsymbol{\theta} = \int_{\boldsymbol{\theta}} p(x_t^{(i)}, \boldsymbol{\theta} | D)  d\boldsymbol{\theta} = p(x_t^{(i)} | D)$, where the last step is marginalization.
>
> W3dcc. eq 15 - please answer W3da first, I will come back to this, hopefully being able to be more precise once you answer W3da. My suspicion is that the authors did not mean the probability distribution over $Z_{\boldsymbol{\theta},t}$ conditioned on $Z_t$, because I do not see a reason why $Z_{\boldsymbol{\theta},t}$ would be stochastic. What causes/drives the stochasticity in $Z_{\boldsymbol{\theta},t}$?
>
> W3dcd. eq 16 - can the authors explain more explicitly by what argument the equalities in this equation hold? They are not an obvious result of standard mathematical rules for manipulating probability distributions, and hence the burden of the proof/derivation lies on the authors, not the reviewer.
>
> W3dce. eq 19 - I second Reviewer-DB7j's point:
> > Eq 19 doesn’t make sense — it only considers the kth cluster, and doesn’t indicate how we combine multiple clusters. As a result, I am unsure as to what the authors’ estimator actually is, and how it combines the energies/log likelihoods for the individual clusters.

---

> > ### Author Response · Authors · 2025-11-13
> >
> > Thank you for the clarification. We now understand your concerns regarding the “W3dc” issue. The main point of disagreement appears to stem from your observation that the marginal distribution
> > $p(x^{(i)}_{t} | \mathcal{D})$ ,
> >
> > is not equivalent to the joint distribution $p(x^{(i)}_{t}, \mathcal{D})$.
> >
> > In the paper, the joint distribution is directly employed because \mathcal{D} is given and fixed, where p(\mathcal{D}) is treated as a constant that infinitely approaches 1.
> >
> > ---
> >
> > Regarding the "W3de", we do not agree with the interpretation that it “only considers cluster k.” This is because, during computation, the term $\frac{1}{n}$ is applied, meaning that samples from other clusters contribute to the normalization of the reliability score.  This is analogous to how confidence scores, though expressed as the maximum class probability, implicitly account for the influence of other classes in the overall distribution.
> >
> > ---
> >
> > Regarding W3dc, we appreciate the reviewer's time in providing clarification to prevent potential misunderstandings in our paper. As for W3de, we respect the reviewer's opinion but do not agree with it.

---

### Meta-Review · Area_Chair_9wqS · 2025-12-25

**Summary:**

* The paper should add more models and datasets to the experiment to make it solid.
* The math derivations should be improved.
* There are many experimental details left open.

**Reviewer Concerns:**

* The authors explained the math issues in the rebuttal.
* The required more experiments are still left open.

**Reviewer Scores:**

all may remain their score

---

### Decision · Program_Chairs · 2026-01-26

Reject